# EXPANDING THE REACH OF FEDERATED LEARNING BY REDUCING CLIENT RESOURCE REQUIREMENTS

## ABSTRACT

Communication on heterogeneous edge networks is a fundamental bottleneck in Federated Learning (FL), restricting both model capacity and user participation. To address this issue, we introduce two novel strategies to reduce communication costs: (1) the use of lossy compression on the global model sent server-to-client; and (2) Federated Dropout, which allows users to efficiently train locally on smaller subsets of the global model and also provides a reduction in both client-to-server communication and local computation. We empirically show that these strategies, combined with existing compression approaches for client-to-server communication, collectively provide up to a $14\times$ reduction in server-to-client communication, a $1.7\times$ reduction in local computation, and a $28\times$ reduction in upload communication, all without degrading the quality of the final model. We thus comprehensively reduce FL's impact on client device resources, allowing higher capacity models to be trained, and a more diverse set of users to be reached.

## 1 INTRODUCTION

Federated Learning (FL) allows users to reap the benefits of models trained from rich yet sensitive data captured by their mobile devices, without the need to centrally store such data (McMahan et al., 2017; Konečný et al., 2016a; Smith et al., 2017). Under the FL paradigm, each device performs training on samples available locally and only communicates intermediate model updates.

Network speed and number of nodes are two of the core systems aspects that differentiate FL from traditional distributed learning in data centers, with network bandwidth being potentially orders of magnitude slower and the number of worker nodes orders of magnitude larger. Together, these issues exacerbate the communication bottlenecks usually associated with distributed learning, increasing both the number of stragglers and the probability of devices dropping out altogether. The problem is further aggravated when working with high capacity models with large numbers of parameters.

Insisting on training these large models using existing federated optimization methods can lead to the systematic exclusion of clients with restricted bandwidth or limited network access from the training stage, and thus to a degraded user experience once these models are served. One naive solution involves training low capacity models with smaller communication footprints, at the expense of model accuracy. As a middle ground, we could develop strategies to reduce the communication footprint of larger, high-capacity models. Recent work (Konečný et al., 2016b) has in fact taken this approach, but only in the context of client-to-server FL communication. Their success with lossy compression strategies is perhaps not surprising, as the clients' lossy, yet unbiased, updates are eventually averaged over many users. However, server-to-client exchanges do not benefit from such averaging. As such, they remain a main bottleneck in our goal of expanding FL's reach.

In this work, we propose two novel strategies to mitigate the server-to-client communication footprint, and empirically demonstrate their efficacy and seamless integration with existing client-to-server strategies. The specific contributions of this paper are as follows:

1. We study lossily compressing the models downloaded by the clients, thus addressing the open question as to whether these approaches are amenable in the context of server-to-client exchanges. We also introduce the use of the theoretically motivated Kashin's representation to reduce the error associated with the lossy compression (Lyubarskii & Vershynin, 2010; Kashin, 1977).

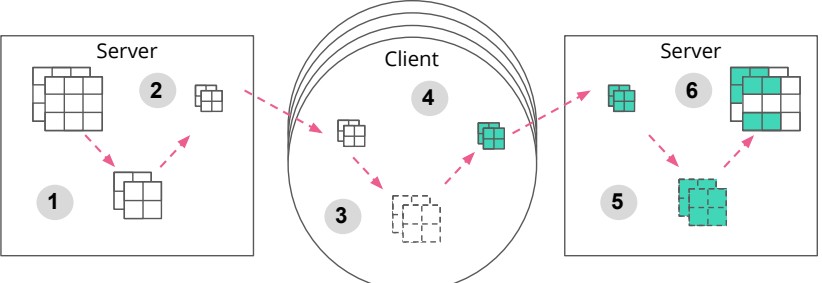

Figure 1: Combination of our proposed strategies during FL training. We reduce the size of the model to be communicated by (1) constructing a sub-model via *Federated Dropout*, and by (2) lossily compressing the resulting object. This compressed model is then sent to the client, who (3) decompresses and trains it using local data, and (4) compresses the final update. This update is sent back to the server, where it is (5) decompressed and finally, (6) aggregated into the global model.

2. We introduce *Federated Dropout*, a technique that builds upon the popular idea of dropout (Srivastava et al., 2014), yet is primarily motivated by systems-related concerns. Our approach enables each device to locally operate on a *smaller* sub-model (i.e. with smaller weight matrices) while still providing updates that can be applied to the larger global model on the server. It thus reduces communication costs by allowing for these smaller sub-models to be exchanged between server and clients, while also reducing the computational cost of local training.

3. We empirically show that not only are these approaches compatible with one another, but with existing client-to-server compression. Combining these approaches during FL training (see Figure 1) reduces the size of the downloaded models up to $14\times$, the size of the corresponding updates up to $28\times$, and the required local computations by up to $1.7\times$, all without degrading the model's accuracy and only at the expense of a slightly slower convergence rate (in terms of number of communication rounds).

## 2 RELATED WORK

We review the relevant related work given our objective of reducing the communication footprint in server-to-client exchanges in Federated Learning (FL).

**Federated Learning** Federated Learning (FL) is a technique that aims to learn a global model over data distributed across multiple edge devices (usually mobile phones) without the data ever leaving the device on which it was generated (McMahan et al., 2017). It brings along a set of statistical (non-IID, unbalanced data) and systems (stragglers, communication bottlenecks, etc.) challenges which differentiate it from traditional distributed learning in the data center, and which have been tackled by several works. For instance, McMahan et al. (2017) propose Federated Averaging (FedAvg), which in its canonical form works by (1) sending the global model to a subset of the available devices, (2) training the model on each device using the available local data, and (3) averaging the local updates to thus end a round of training. In contrast Smith et al. (2017) present a multi-task variant that also models the relationship between clients in order to learn personalized yet related models for each device. Nonetheless, all approaches we are aware of (including the two aforementioned ones) require continued exchanges between a central server and its clients across a potentially slow network.

**Communication-efficient distributed learning** Distributed learning is known to suffer from communication overheads associated with the frequent gradient updates exchanged among nodes (Wang et al., 2018; Dean et al., 2012; Smith et al., 2018; Reddi et al., 2016). To reduce these bottlenecks, recent studies focus on communicating a sparsified, quantized or randomly subsampled version of the updates. Although these operations introduce noise, they have been shown both empirically and theoretically to maintain the quality of the trained models. We refer the reader to the introduction of Wang et al. (2018) for more details and references.

In the context of FL, Konečný et al. (2016b) successfully perform lossy compression on the client-to-server exchanges (i.e. the model updates). Of particular interest is their use of the randomized Hadamard transform to reduce the error incurred by the subsequent quantization. This is due to the fact that the Hadamard transform, in expectation, spreads a vector's information more evenly across its components (Suresh et al., 2017; Konečný & Richtárik, 2016).

We note, however, that neither the work on traditional distributed learning nor the work of Konečný et al. (2016b) considers compressing the server-to-client exchanges. Nevertheless, in FL, downloading a large model can still be a considerable burden for users, particularly for those in regions with network constraints. Furthermore, as FL is expected to deal with a large number of devices, communicating the global model may even become a bottleneck for the server (as it would, ideally, send the model to the clients in parallel).

**Model compression**    Deep models tend to demand significant computational resources both for training and inference. Using them on edge devices is therefore not a straightforward task. Because of this, several recent works have proposed compressing the models before deploying them on-device (Ravi, 2018). Popular alternatives include pruning the least useful connections in a network (Han et al., 2016; 2015), weight quantization (Hubara et al., 2016; Lin et al., 2017; De Sa et al., 2018), and model distillation (Hinton et al., 2015). Many of these approaches, however, are not applicable for the problems addressed in this work, as they are either ingrained in the training procedure (and our server holds no data and performs no actual training) or are mostly optimized for inference. In the context of FL, we need something computationally light that can be efficiently applied in every round and that also allows for subsequent local training. We do note, however, that some of the previously mentioned approaches could potentially be leveraged at inference time in the federated setting, and exploring these directions would be an interesting avenue for further research.

## 3    METHODS

In this section, we present our proposed strategies for reducing Federated Learning's (FL) server-to-client communication costs, namely lossy compression techniques (Section 3.1) and *Federated Dropout* (Section 3.2). We introduce the strategies separately, but they are fully compatible with one another (as we show in Section 4.4).

### 3.1    LOSSY COMPRESSION

Our first approach at reducing bandwidth usage consists of using lightweight lossy compression techniques that can be applied to an already trained model and that, when reversed (i.e. after decompression), maintain the model's quality. The particular set of techniques we propose are inspired by those successfully used by Konečný et al. (2016b) to compress the client-to-server updates. We apply them, however, to the server-to-client exchanges, meaning we do not get the benefit of averaging the noisy decompressions over many updates.

Our method works as follows: we reshape each to-be-compressed weight matrix in our model into a vector $w$ and (1) apply a basis transform to it. We then (2) subsample and (3) quantize the resulting vector and finally send it through the network. Once received, we simply execute the respective inverse transformations to finally obtain a noisy version of $w$.

**Basis transform**    Previous work (Lyubarskii & Vershynin, 2010; Konečný et al., 2016b) has explored the idea of using a basis transform to reduce the error that will later be incurred by perturbations such as quantization. In particular, Konečný et al. (2016b) use the random Hadamard transform to more evenly spread out a vector's information among its dimensions. We go even further and also apply the classical results of Kashin (1977) to spread a vector's information *as much as possible* in every dimension (Lyubarskii & Vershynin, 2010). Thus, Kashin's representation mitigates the error incurred by subsequent quantization compared to using the random Hadamard transform. For a more detailed discussion, we refer the reader to Section A.3 in the Appendix.

**Subsampling**    For $s \in [0, 1)$, we zero out a $1 - s$ fraction of the elements in each weight matrix, appropriately re-scaling the remaining values. The elements to zero out are picked uniformly at random. Thus, we only communicate the non-zero values and a random seed which allows recovery of the corresponding indices.

(i) Original network, with $a_1$, $b_2$, and $c_3$ marked for dropout

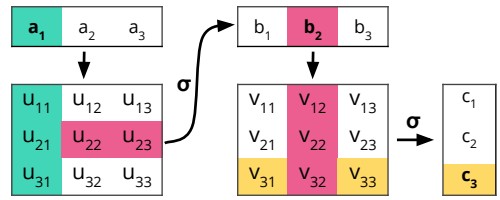

(ii) On-device network after Federated Dropout

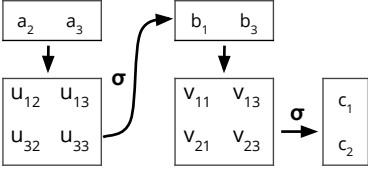

Figure 2: *Federated Dropout* applied to two fully-connected layers. Notices activation vectors $a, b = \sigma(Ua)$ and $c = \sigma(Vb)$ in (I). In this example, we randomly select exactly one activation from each layer to drop, namely $a_1$, $b_2$, and $c_3$, producing a sub-model with $2 \times 2$ dense matrices, as in (II).

**Probabilistic quantization** For a vector $\boldsymbol{w} = (w_1, \ldots, w_n)$, let us denote $w_{\min} = \min_j \{w_j\}_{j=1}^n$ and $w_{\max} = \max_j \{w_j\}_{j=1}^n$. Uniform probabilistic 1-bit quantization replaces every element $w_i$ by $w_{\min}$ with probability $\frac{w_{\max} - w_i}{w_{\max} - w_{\min}}$, and by $w_{\max}$ otherwise. It is straightforward to verify this yields an unbiased estimate of $\boldsymbol{w}$. Now, for $q$-bit uniform quantization, we first equally divide $[w_{\min}, w_{\max}]$ into $2^q$ intervals. If $w_i$ falls in the interval bounded by $w'$ and $w''$, the quantization operates by replacing $w_{\min}$ and $w_{\max}$ in step two of the above algorithm by $w'$ and $w''$, respectively.

### 3.2 FEDERATED DROPOUT

To further reduce communication costs, we propose an algorithm in which each client, instead of locally training an update to the whole global model, trains an update to a smaller sub-model. These sub-models are subsets of the global model and, as such, the computed local updates have a natural interpretation as updates to the larger global model. We call this technique *Federated Dropout* as it is inspired by the well known idea of dropout (Srivastava et al., 2014), albeit motivated primarily by systems-level concerns rather than as a strategy for regularization.

In traditional dropout, hidden units are multiplied by a random binary mask in order to drop an expected fraction of neurons during each training pass through the network. Because the mask changes in each pass, each pass is effectively computing a gradient with respect to a different sub-model. These sub-models can have different sizes (architectures) depending on how many neurons are dropped in each layer. Now, even though some units are dropped, in all implementations we are aware of, activations are still multiplied with the original weight matrices, they just have some useless rows and columns.

To extend this idea to FL and realize communication and computation savings, we instead zero out a *fixed* number of activations at each fully-connected layer, so all possible sub-models have the same reduced architecture; see Figure 2. The server can map the necessary values into this reduced architecture, meaning only the necessary coefficients are transmitted to the client, re-packed as smaller dense matrices. The client (which may be fully unaware of the original model's architecture) trains its sub-model and sends its update, which the server then maps back to the global model[1]. For convolutional layers, zeroing out activations would not realize any space savings, so we instead drop out a fixed percentage of filters.

This technique brings two additional benefits beyond savings in server-to-client communication. First, the size of the client-to-server updates is also reduced. Second, the local training procedure now requires a smaller number of FLOPS per gradient evaluation, either because all matrix-multiplies are now of smaller dimensions (for fully-connected layers) or because less filters have to be applied (for convolutional ones). Thus, we reduce local computational costs.

## 4 EXPERIMENTAL RESULTS

In this section, we first present our experimental setup (Section 4.1) before presenting results for our lossy compression (Section 4.2) and *Federated Dropout* (Section 4.3) strategies. Finally, we

---

[1]This can be done by communicating a single random seed to the client and back, or via state on the server.

Table 1: Summary of Datasets used in the experiments.

| Dataset | # of users | IID | Training samples per user | | Test samples per user | |
|---|---|---|---|---|---|---|
| | | | mean | $\sigma$ | mean | $\sigma$ |
| MNIST | 100 | Yes | 600 | 0 | 100 | 0 |
| CIFAR-10 | 100 | Yes | 500 | 0 | 100 | 0 |
| EMNIST | 3550 | No | 181.46 | 71.15 | 45.37 | 17.79 |

show experiments that use both of these strategies in tandem with those proposed in Konečný et al. (2016b) to also compress client-to-server exchanges (Section 4.4).

## 4.1 EXPERIMENTAL SETUP

**Optimization Algorithm** We focus on testing our strategies against already established FL benchmarks. In particular, we restrict our experiments to the use of Federated Averaging (FedAvg) (McMahan et al., 2017).

**Datasets** We use three datasets in our experiments: MNIST (LeCun et al., 1998), CIFAR-10 (Krizhevsky & Hinton, 2009) and Extended MNIST or EMNIST (Cohen et al., 2017). The first two were used to benchmark the performance of FedAvg and of lossy compression for client-to-server updates (Konečný et al., 2016b). For these two datasets, we use the artificial IID partition proposed by these previous works. Meanwhile, EMNIST is a dataset that has only recently been introduced as a useful benchmark for FL. Derived from the same source as MNIST, it also includes the identifier of the user that wrote the character (digit, lower or upper case letter), creating a natural and much more realistic partition of the data. Table 1 summarizes the basic dataset properties. Due to space constraints, we relegate the MNIST results to Appendix B, though all conclusions presented here also qualitatively hold for these experiments.

**Models** For MNIST's digit recognition task we use the same model as McMahan et al. (2017): a CNN with two 5x5 convolution layers (the first with 32 channels, the second with 64, each followed by 2x2 max pooling), a fully connected layer with 512 units and ReLu activation, and a final softmax output layer, for a total of more than $10^6$ parameters. For CIFAR-10, we use the all convolutional model taken from what is described as "Model C" in Springenberg et al. (2015), which also has a total of over $10^6$ parameters. Finally, for EMNIST we use a variant of the MNIST model with 2048 units in the final fully connected layer. While none of these models is the state-of-the-art, they are sufficient for evaluating our methods, as we wish to measure accuracy degradation against a baseline and not to achieve the best possible accuracy on these tasks.

**Hyperparameters** We do not optimize our experiments for FedAvg's hyperparameters, always using those that proved to work reasonably well in our baseline setting which involves no compression and no *Federated Dropout*. For local training at each client we use static learning rates of 0.15 for MNIST, 0.05 for CIFAR-10 and 0.035 for EMNIST. We select 10 random clients per round for MNIST and CIFAR-10, and 35 for EMNIST. Finally, each selected client trains for one epoch per round using a batch size of 10.

## 4.2 LOSSY COMPRESSION

We focus on testing how the compression strategies presented in Section 3.1 impact the global model's accuracy. Like Konečný et al. (2016b), we don't compress all variables of our models. As they mention, compressing smaller variables causes significant accuracy degradation but translates into minuscule communication savings. As such, we don't compress biases for any of the models[2].

In our experiments, we vary three parameters:

1. The type of basis transform applied: no transform or identity (I), randomized Hadamard transform (HD) and Kashin's representation (K).

---

[2]Unlike Konečný et al. (2016b), we do compress all 9 convolutional layers in the CIFAR-10 model, not just the 7 in the middle.

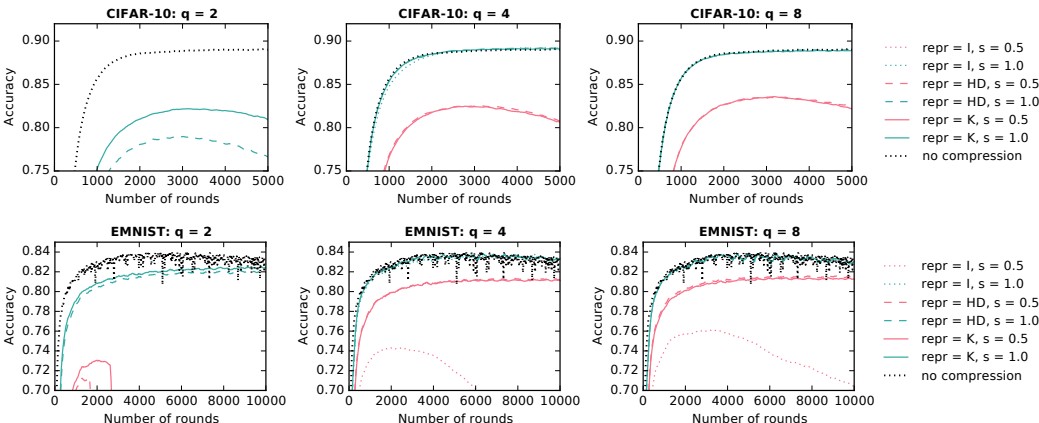

Figure 3: Effect of varying our lossy compression parameters on CIFAR-10 and EMNIST.

2. The subsampling rate $s$, which refers to the fraction of weights that are kept (i.e. $1 - s$ of the weights are zeroed out).
3. The number of quantization bits $q$.

Figure 3 shows the effect of varying these parameters for CIFAR-10 and EMNIST. We repeat each experiment 10 times and report the mean accuracy among these repetitions. The three main take-aways from these experiments are: (1) for every model, we are able find a setting of compression parameters that at the very least matches our baseline; (2) Kashin's representation proves to be most useful for aggressive quantization values; and (3) it appears that subsampling is not all that helpful in the server-to-client setting. We proceed to give more details about these highlights.

The first takeaway is that, for every model, we are indeed able find a setting of compression parameters that matches or, in some cases, slightly outperforms our baseline. In particular, we are able to quantize every model to $4$ bits, which translates to a reduction in communication of nearly $8\times$.

The second takeaway is that Kashin's representation proves to be most useful for aggressive quantization values, i.e. for low values of $q$. In our experiments, gains were observed only in regimes where the overall accuracy had already degraded, but we hypothesize that the use of Kashin's representation may provide clearer benefits in the compression of client-to-server gradient updates, where more aggressive quantization is admissible. We also highlight that using Kashin's representation may be beneficial for other datasets. Indeed, its computational costs are comparable to that of the random Hadamard transform while also providing better theoretical error rates (see Section A.1). We refer the reader to Section A.3 in the Appendix, where we show preliminary results that demonstrate Kashin's potential to dominate over the randomized Hadamard transform in compressing fully-trained models, particularly for small values of $q$.

Finally, it appears that subsampling is not all that helpful in this server-to-client setting. This contrasts with the results presented by Konečný et al. (2016b) for compressed client-to-server updates, where aggressive values of $s$ were admissible. This trend extends to the other compression parameters: server-to-client compression of global models requires much more conservative settings than client-to-server compression of model updates. For example, for CIFAR-10, Konečný et al. (2016b) get away with using $s = 0.25$ and $q = 8$ under a random Hadamard transform representation[3]. Meanwhile, in Figure 3 we can see that, for the same $q$ and representation, $s = 0.5$ already causes an unacceptable degradation of the accuracy. This is not surprising, since it is expected that the updates' error will cancel out once several of them get aggregated at the server, which is not true for model downloads.

---

[3]The updates for CIFAR-10 can actually be compressed up to 2 bits.

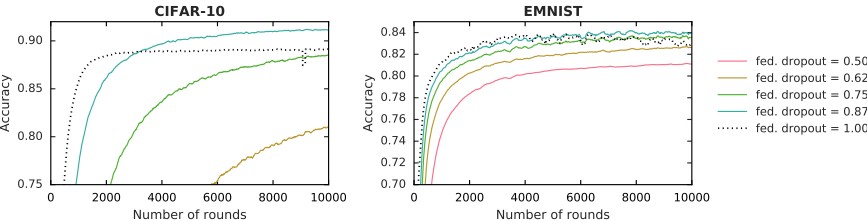

Figure 4: Results for *Federated Dropout*, varying the percentage of neurons *kept* in each layer.

## 4.3 FEDERATED DROPOUT

We focus on testing how the global model's accuracy deteriorates once we use the strategy proposed in Section 3.2. In these experiments, we vary the percentage of neurons (or filters for the case of convolutional layers) that are *kept* on each layer of our models (we call this the federated dropout rate). We always keep the totality of the input and logits layers, and never drop the neuron that can be associated to the bias term.

Figure 4 shows how the convergence of our three models behaves under different federated dropout rates. We repeat each experiment 10 times and report the mean among these repetitions. The main takeaway from these experiments is that, for every model, it is possible to find a federated dropout rate less than $1.0$ that matches or, in some cases, even improves on the final accuracy of the model.

A federated dropout rate of $0.75$ seems to work across the board. This corresponds to dropping $25\%$ of the rows and columns of the weight matrices of fully-connected layers (which translates to a $\sim 43\%$ reduction in size), and to dropping the same percentage of filters of each convolutional layer. Now, because fully connected layers correspond to most of the parameters of the MNIST and EMNIST models, the $\sim 43\%$ reduction will apply to them both in terms of the amount of data that has to be communicated and of the number of FLOPS required for local training. Meanwhile, because our CIFAR model is fully convolutional, gains will be of $25\%$.

As a final comment, we note that more aggressive federated dropout rates tend to slow down the convergence rate of the model, even if they sometimes result in a higher accuracy.

## 4.4 REDUCING THE OVERALL COMMUNICATION COST

Our final set of experiments shows how our models behave once we combine our two strategies, lossy compression and *Federated Dropout*, with existing client-to-server compression schemes (Konečný et al., 2016b), in order to explore how the different components of this end-to-end, communication efficient framework interact. To do this, we evaluate how our models behave under 3 different compression schemes (aggressive, moderate and conservative) and 4 different federated dropout rates ($0.5$, $0.625$, $0.75$ and $0.875$). The values for these schemes and rates were picked based on the observed behavior during the previous experiments, being somewhat more conservative as we are now combining different sources of noise. Table 2 describes the settings for each scheme.

Figure 5 shows how our CIFAR-10 and EMNIST models behave under each of the previously mentioned conditions. We repeat each experiment 5 times and report the mean among these repetitions. For all three models, a federated dropout rate of $0.75$ resulted in models with no accuracy degradation under all compression schemes except for the most aggressive. For MNIST and EMNIST, this translates into server-to-client communication savings of $14\times$, client-to-server savings of $28\times$ and a reduction of $1.7\times$ in local computation, all without degrading the accuracy of the final global model (and sometimes even improving it). For CIFAR-10, we provide server-to-client communication savings of $10\times$, client-to-server savings of $21\times$ and local computation savings of $1.3\times$.

Based on these results, we also hypothesize that a federated dropout rate of $0.75$ combined with a moderate or conservative compression scheme will be a good starting point when setting these parameters in practice.

Table 2: Settings for each of our proposed compression schemes.

| Scheme | Client-to-Server | | | Server-to-Client | | |
|---|---|---|---|---|---|---|
| | transf. | $s$ | $q$ | transf. | $s$ | $q$ |
| Aggressive | Kashin's | 0.4 | 2 | Kashin's | 1.0 | 3 |
| Moderate | Kashin's | 0.5 | 4 | Kashin's | 1.0 | 5 |
| Conservative | Kashin's | 1.0 | 8 | Kashin's | 1.0 | 8 |

Figure 5: Effect of using both compression and *Federated Dropout* on CIFAR-10 and EMNIST.

## 5 CONCLUSIONS AND OPEN QUESTIONS

The ecosystem currently targeted by Federated Learning (FL) is marked by heterogeneous edge networks that can potentially be orders of magnitude slower than the ones in datacenters. At the same time, FL can be quite demanding in terms of bandwidth, particularly when used to train deep models. We are thus at risk of either restricting the type of models we are able to train using this technique, or of excluding large groups of users from federated training. Both issues are problematic, but because access to high-end networks also appears to be correlated to sensitive factors such as income and age (Anzilotti, 2016; Pew Research Center, 2018), the latter may have implications related to fairness, making it particularly sensitive as we continue the adoption of FL systems.

Our work dramatically reduces the communication overheads in FL by (1) using lossy compression techniques on the server-to-client exchanges and by (2) using *Federated Dropout*, a technique that only communicates subsets of the global model to each client. We empirically show that a combination of our strategies with previous work allows for up to a $14\times$ reduction in server-to-client communication, a $1.7\times$ reduction in local computation and a $28\times$ reduction in client-to-server communication.

In future work, we plan to: explore the efficacy of introducing a server step size in order to account for the use of different sub-models in *Federated Dropout*; investigate the possibility of using the same sub-models for all the selected clients in one round; and further characterize the benefits of Kashin's representation in compressing the gradient updates in FL and in traditional model serving. An additional future direction to pursue related to fairness involves studying the effect of adaptively using these strategies (i.e. using more aggressive compression and federated dropout rates for some users) to prevent unfairly biased models. Finally, we note that the success of *Federated Dropout* suggests an entirely new avenue of research in which smaller, perhaps personalized, sub-models are eventually aggregated into a larger, more complex model that can be managed by the server. Contrary to the classic datacenter setting, the computational overhead associated with first creating and then aggregating the sub-models is justified in FL.

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

# A    Kashin's Representation

For reasons of space, we have relegated a more detailed discussion of Kashin's representation (see Section 3) to the Appendix. In this section, we briefly discuss Kashin's representation both from a theoretical (Section A.1) and practical (Section A.2) standpoints. Finally, we present some preliminary results that argue the potential of Kashin's representation to dominate over the random Hadamard transform with respect to the size vs. accuracy trade-off (Section A.3).

## A.1    Theoretical Overview

The idea of using the classical results of Kashin (1977) to increase the robustness of coefficients to perturbations was first introduced by Lyubarskii & Vershynin (2010). Their result states that, given a tight frame satisfying a form of uncertainty principle, a weaker notion of the RIP (Candes et al., 2006), it is possible to convert the frame representation of every vector into the more robust *Kashin's representation*, whose coefficients will have the smallest possible dynamic range.

**Error rates**    Since the results of Suresh et al. (2017) (who quantified the reduction in quantization error due to the Hadamard transform) rely on exactly this notion of dynamic range, and assuming the subsampled randomized Hadamard transform satisfies the uncertainty principle, Theorem 3.5 of Lyubarskii & Vershynin (2010) can be directly used as a drop-in replacement for Lemma 7 in Suresh et al. (2017), removing the logarithmic dependence on dimension from Theorem 3 therein, matching the lower bounds. We do not provide the complete proof as, beyond drawing this connection, it does not imply any novelty whatsoever. However, an open question remains, as we are not aware of a result showing what are the parameters of the uncertainty principle guaranteed by the subsampled randomized Hadamard transform. They exist however, as the transform is known to satisfy the RIP (Foucart & Rauhut, 2013), which is a stronger notion.

## A.2    Practical Considerations

In practice, given a tight frame, the algorithm for computing Kashin's representation is straightforward. It runs for $n$ iterations, and takes parameters $\eta, \delta$ as input. In a single iteration, one first computes the frame coefficients, projects them onto a $L_\infty$ ball, and reconstructs the error in the original domain. Another iteration proceeds starting with the reconstructed error and a smaller ball. We refer the reader to Lyubarskii & Vershynin (2010) for more details regarding $\eta, \delta$ and their relationship with the uncertainty principle.

In our work, we use the randomized Hadamard transform as the initial tight frame (see Section A.1 for details on why this is possible). We also run the algorithm for just $n = 2$ iterations (as very often this provides most of the benefit), fixed $\delta = 1$, and used a variant of the algorithm which yields an exact representation (omitting the $L_\infty$ projection in the last iteration). Given this, the choice of $\eta$ is irrelevant. The dominant part of the computation is then three applications of the fast Walsh-Hadamard transform, as opposed to a single one in Konečný et al. (2016b)).

As a particular example, say we are to compress an 80-dimensional vector. We first pad the vector with zeros, so that its dimension is 128 (the closest larger power of 2). Then, we multiply the vector by a diagonal matrix with independent Rademacher random variables ($D \in \mathbb{R}^{128 \times 128}$), followed by the application of the fast Walsh-Hadamard transform ($H \in \mathbb{R}^{128 \times 128}$). The first 80 columns of the matrix $HD$ correspond to the tight frame used to find the Kashin's representation. Nonetheless, we avoid representing this explicitly.

Finally, note that, if the initial dimension was a power of 2, we need to pad zeros to the next power of 2 in order to realize any benefit over just using the Hadamard transform.

## A.3    Dominance over Hadamard

Given the theoretical properties of Kashin's representation, we hypothesize it should dominate the random Hadamard transform when it comes to the size vs. accuracy trade-off. A preliminary experiment to corroborate this hypothesis is the following:

1. We train an MNIST model until we get an accuracy of around 99.3%.

2. We compress the original model using some linear transform, some subsampling ratio and some number of quantization bits.
3. We decompress the model and evaluate both its new accuracy and its $L_2$ distance to the original model.
4. We repeat the previous two steps for different linear transforms (identity, random Hadamard transform and Kashin's representation), subsampling ratios (0.25, 0.5 and 1.0) and quantization bits (1, 2, 4, 8, 16).

An important detail is that, whenever we use Kashin's representation, we do a grid search over the best values for $n$ (from 1 to 10) and $\eta$. However, $\delta$ is kept fixed as 1.

The results of this experiment are shown in Figure 6. In the legend, R corresponds to rotation — I for identity, HD for randomized Hadamard, Kashin for Kashin based on the randomized Hadamard; and SR corresponds to subsampling ratio — the fraction of elements to be kept non-zero. In the top row, the figure shows the relationship of the accuracy of the compressed model vs. the number of bits used for quantization, and vs. the model's size (in MB). In the bottom row, the $L_2$ error incurred is plotted against the same. It is very clear then that Kashin's representation does dominate the other two representations when it comes to the size vs. accuracy trade-off, making up the Pareto frontier for all combinations of subsampling ratio and quantization bits. Nevertheless, we did optimize over the parameters associated with Kashin's algorithm, something that does not need to be done for the random Hadamard transform. In Section A.2, we propose a set of values that worked well enough for our experiments, but further exploration on how to easily determine these values is in order.

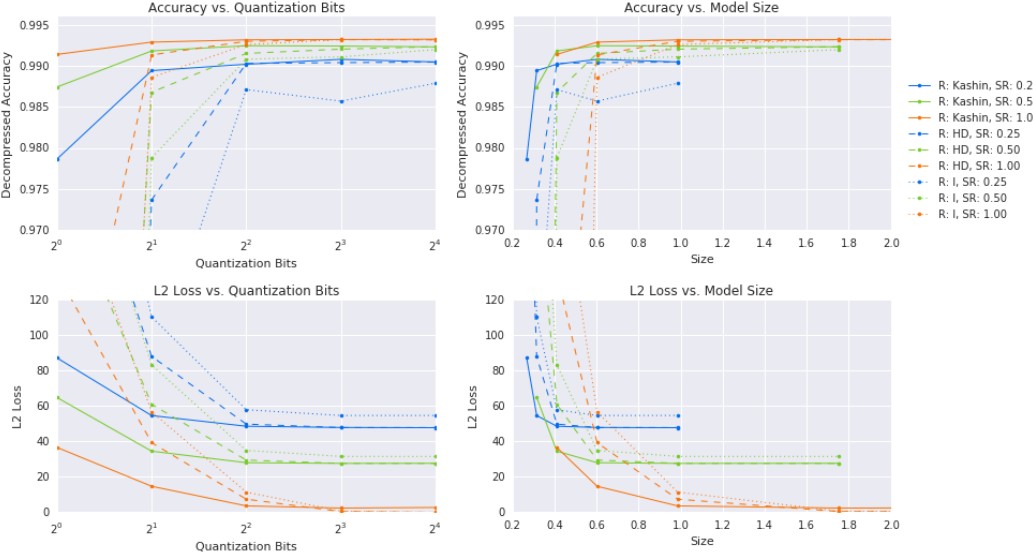

Figure 6: Compressing an already trained MNIST model with linear transform + subsampling + uniform quantization.

# B  MNIST EXPERIMENTAL RESULTS

For reasons of space, we have relegated the experimental results using MNIST (see Section 4) (Section A) to the Appendix.

Figure 7 shows the results of using our lossy compression on MNIST under the experimental setup presented in Section 4.2. Meanwhile, Figure 8 shows the results of using *Federated Dropout* (see Section 4.3 for details). Finally, Figure 9 shows the results of performing both lossy compression for downloads and uploads, as well as *Federated Dropout*, as described in Section 4.4.

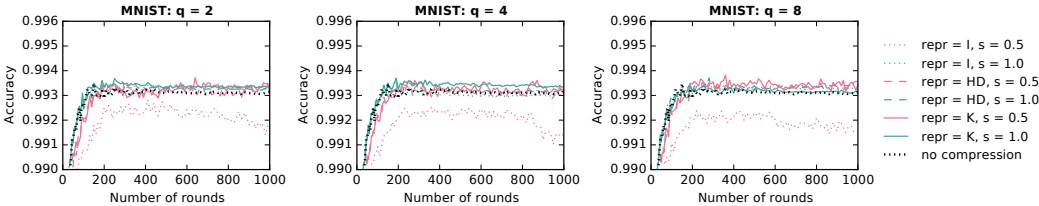

Figure 7: Effect of varying our lossy compression parameters on the convergence MNIST.

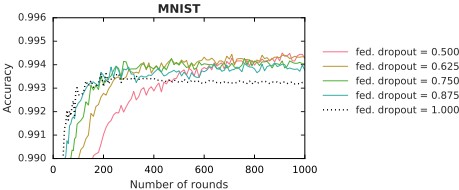

Figure 8: Effect of varying the percentage of neurons *kept* in each layer on MNIST.

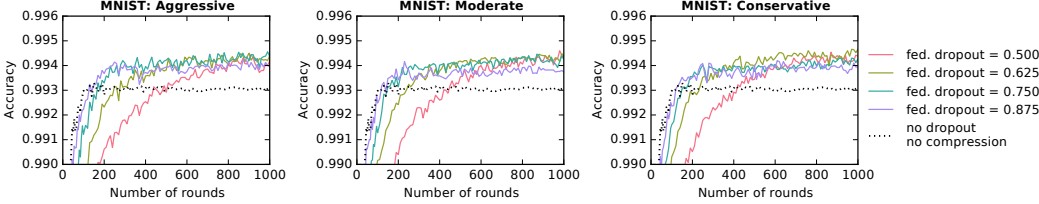

Figure 9: Effect of using both lossy compression and *Federated Dropout* on MNIST.

