# OpenReview forum: "Expanding the Reach of Federated Learning by Reducing Client Resource Requirements"
_ICLR.cc/2019/Conference_

### Official Review · AnonReviewer1 · 2018-11-03
**This paper focuses on lossy compression techniques and federated dropout strategies to control the update burden that’s needed to coordinate nodes in a federated learning setting.**

**Rating:** 5
**Confidence:** 4

**Review:**

The paper is well written and addresses an interesting problem. Overall, I do find the federated dropout idea quite interesting. As for the lossy compression part, I am a bit skeptical on its application for this problem. In general, I believe that the manuscript could greatly benefit from answering the questions that I am raising below. It would certainly help me better appreciate the contributions of this work.

The lossy aspect of the compression inevitably introduces performance downgrades. However, compression/communication systems are designed to make sure that the information dropped is not important for the task at hand (e.g., high frequencies that are not perceived by our eyes in the spatial domain are typically dropped when compressing images through zig zag scanning after transformation). Randomly dropping coefficients as suggested in this paper seems odd to me (the subsampling technique that is used). Can you justify this approach? The manuscript does hint that this approach provides lukewarm results. Could there be a better approach that focuses on parts of the model that deemed “less” important if a notion of coefficient importance can be derived?

Can you emphasize more the benefits of compression and federated drop out, versus training a low capacity model with less parameters? The introduction refers to the low capacity approach as a naive model. Could this be compared experimentally? This would help better appreciate the benefits of the federated dropout strategies that are proposed here. In the experiments, could you explain why increases in q (quantization steps) seems to lead to limited or marginal accuracy improvements?

For the results shown in Figure 4, did you also use any form of subsampling and quantization? Also, do you have a justification for why with some amounts of dropout, the accuracy may improve but at a slower pace (pretty much the punch line of these experiments)? It is an interesting finding but it is counter intuitive and requires explanations in my view.

On the communication cost experiments, can you explain precisely how did you compute these reduction factors? Did you tolerate some form of accuracy degradation? Also, did you consider the fact that more "rounds" are needed to get to a target accuracy level? Is there a cost associated with these additional rounds and was that cost taken into consideration? Adding clarity on this would certainly help.

---

> ### Author Response · Authors · 2018-11-19
> **Thank you for your constructive feedback. Please find answers to the specific concerns below (Part 2)**
>
> - “...why with some amounts of dropout, the accuracy may improve but at a slower pace?”
> Answer: This is in line with the empirical observations of (standard) dropout. We only have weak suggestions for why this might be the case, which will require more work to support: The effect we see might be because the approach effectively creates a random ensemble of models within the single global model. Moreover, it might be possible to get a speed up stemming from the following observation: Since we generate an update for only a subset of the model parameters, we might be able to utilize a smarter server averaging scheme - instead of simply averaging the updates as done currently. Investigating this might be an interesting follow-up work.
>
> - “On the communication cost experiments, can you explain precisely how did you compute these reduction factors?”
> Answer: The reduction factors do not tolerate any form of accuracy degradation, and are calculated from the client’s perspective. In particular, the presented reduction factors are computed from the “Moderate” compression scheme presented in Table 2: the 9.6x reduction in server-to-client communication is the compounding of an 6.4x reduction due to quantization (to 5 bits) and a 1.5x reduction due to federated dropout (rate of 0.8 corresponding to ~0.8*0.8 factor of saving); the 1.5x reduction in local computation is due to federated dropout (rate of 0.8). The 24x reduction in upload communication is the compounding of a 16x reduction due to quantization (4 bits) and subsampling (s = 0.5), and a 1.5x reduction due to federated dropout (rate of 0.8). However, notice that, with the addition of dropout for convolutional layers, these reductions changed (improved) slightly (see note to all reviewers). We have updated the numbers in our submission.
>
> - “...did you consider the fact that more "rounds" are needed to get to a target accuracy level?”
> Answer: In practice, using compression and Federated Dropout will make the rounds complete faster. Thus, without access to an actual production deployment, it is generally impossible to say what will best in terms of runtime. Therefore, we think the number of rounds is the best “fair” comparison. At the same time, note that slightly longer runtime would be a welcome price to pay for higher final accuracy. We see this point is not clear in the paper and we will add a remark on this.

---

> ### Author Response · Authors · 2018-11-19
> **Thank you for your constructive feedback. Please find answers to the specific concerns below (Part 1)**
>
> Thank you for your feedback, helping us see which parts of our contributions are not getting across clearly enough. Please find answers to the specific concerns below:
>
> - “Randomly dropping coefficients as suggested in this paper seems odd to me...”
> Answer: Note the following two aspects of this technique: a) we propose to use it jointly with a random basis transform, which spreads the subsampling and quantization impact randomly throughout the whole input domain; and b) it has been shown to be practically effective for Client-to-Server communication (as we mention in the paper). We proceed to expand further on these two points:
> a) If we apply the subsampling in a different domain (such as the ones obtained by applying the Hadamard transform or the Kashin representation), the loss of information in one coefficient is reflected as a random noise spread throughout the original domain. Subsampling multiple coefficients will produce a similar random effect for every coefficient in the input domain, thus reducing the overall loss of information. The error incurred by quantization is similarly reduced.
> b) We believe it is useful for the community to also share negative results. Even though subsampling is not effective in Figure 3 (using it for only Server-to-Client compression), in Figure 5, the best result is indeed obtained when subsampling the Client-to-Server updates. It is in general a rather aggressive method, and our practical experience was the following: If we already are in relatively aggressive quantization regime (say, q=4), and we want to reduce the representation size by another factor of 2, we have two options: we can either make quantization very aggressive (change q from 4 to 2), or we can add some subsampling (change s from 1 to 0.5). The latter usually leads to smaller additional error, and it is the setting presented in Figure 5.
>
> As suggested, we could also focus on adaptive approaches that try to subsample somehow “less important” coefficients. The downside is that we would need to communicate both values and their corresponding indices (as opposed to values and a shared random seed for data independent subsampling). We did try a preliminary experiment, where we used variable length coding to realize full representation savings, but found it overall less effective, and thus did not perform full experiments. This was particularly true when using Kashin’s representation. Because this representation spreads the information in a vector much more uniformly, any adaptive scheme has smaller potential for improvement.
>
> - “Can you emphasize more the benefits of compression and federated drop out, versus training a low capacity model with less parameters?”
> Answer: Note that the technique we propose is independent of the model being trained. Hence, if one wanted to use a smaller model to solve a particular task, our method would further optimize the efficiency (in terms of communication and local computation) of that model. The contribution should thus be rather seen as follows: if we are interested in the tradeoff between model size and overall computational requirements, our proposal shifts the tradeoff curve to strictly better possibilities. The remark in the introduction is highlighting that, in a resource constrained environment, instead of only training a smaller capacity model, our proposed method *enables* more complex models to be trained.
>
> - “...why increases in q (quantization steps) seems to lead to limited or marginal accuracy improvements?”
> Answer: A large q is close to the baseline - full floating precision. This is thus the expected behavior. The interesting question we are exploring is the opposite - how much can we *decrease* q, before we see an impact on the overall accuracy?
>
> - “Figure 4 - any subsampling or quantization?”
> Answer: No, the experiment in Fig 4 explores only the effect of Federated Dropout without other changes. The combination of all proposed ideas is in Figure 5.

---

### Official Review · AnonReviewer3 · 2018-11-05
**The paper adress the ressource issue of federated learning by introducing a lossy compression on the global model and what they coin a Federated Dropout. While not completely familiar with compression schemes, I saw a couple of statements requiring formal support.**

**Rating:** 5
**Confidence:** 3

**Review:**

The paper tackles a major issue in distributed learning in general (and not only the federated scheme), which is communication bottleneck.

I am not fully qualified to judge and would rather listen to the opinion of more qualified reviewers, I was annoyed by some aspects of the paper:

1) many claims required formal support (proofs), as an example: "more aggressive dropout rates ted to slow down the convergence rate of the model, even if they sometimes result in a higher accuracy" is a statement that would benefit from analyzing the dropout out effect on convergence, something that wouldn't be hard to do given the extensive theoretical toolbox on distributed optimization.

2) no comparison with other compression schemes (see e.g. Alistarh et al.'s ZipML (NIPS or ICML 2017) and followups)

3) proving an unbiased-ness guarantee out of the Probabilistic quantization (section 3.1) would have been a minimal requirement in my opinion.

I encourage the authors to further expand those points, but would happily lighten-up my skepticism if more qualified reviewers say that we do not need such guarantees as the one in point 1 and 3. (the few compression papers I know provide that)

---

> ### Author Response · Authors · 2018-11-19
> **We thank the reviewer for their comments and proceed to address the three points they raised.**
>
> We thank the reviewer for their comments and for highlighting the relevance of our work for the broader distributed learning community. We proceed to address the three points you raised:
>
> 1) The particular observation you mention is in line with previous empirical observations of the effect of (standard) dropout. We don’t analyse this effect, however, as we are not aware of any rigorous argument of why standard dropout works in the first place. We understand dropout as a heuristic that has proven to be incredibly useful and is backed by some interesting intuitions, but not as a principled approach for which we can prove convergence.
>
> 2) The ZipML framework proposes using lower precision at various parts of the training pipeline. Many of these ideas are orthogonal, yet compatible with what we propose. The parts that can be seen as alternatives to our methods (i.e. compressing gradients) are best summarized in algorithms such as QSGD or Terngrad (also called out by another reviewer). We copy our response here:
>
> We did not compare with these for two reasons.
> a) These methods were proposed for compression of gradient updates. In particular, the Terngrad paper argues for using the empirical distributions of the coefficients of such gradients. Even though those arguments would not directly apply to our setting, we could probably still use it for the Client-to-Server compression. However, we do not see a good reason why the proposal would be useful for compressing the state of the model being trained (i.e. Server-to-Client), which is the central concern of our paper.
> b) We performed a series of preliminary experiments where we compressed a variety of random vectors using QSGD and other techniques. The results of these small experiments suggested that in the tradeoff between accuracy and representation size, (I) uniform quantization was dominated by QSGD, and (II) QSGD was in turn dominated by the combination of Kashin’s representation and uniform quantization.
>
> 3) The proof of this is elementary, and we do not want to appear to claim it is a novel insight. We are happy to provide explicit reference to an existing, more general argument, e.g., one in Suresh et al. or in Konecny and Richtarik, both of which we cite.
>
> If you have other concrete comments on what would strengthen the paper, we will be more than happy to incorporate them.

---

### Official Review · AnonReviewer2 · 2018-11-05
**The paper presents some new approaches for communication efficient Federated Learning that allows for training of large models on heterogeneous edge devices.**

**Rating:** 4
**Confidence:** 5

**Review:**

The paper presents some new approaches for communication efficient Federated Learning (FL) that allows for training of large models on heterogeneous edge devices. In FL, heterogeneous edge devices have access to potentially non-iid samples of data points and try to jointly learn a model by averaging their local models at a parameter server (the cloud). As the bandwidth of the up/downlink-link may be limited communication overheads may become the bottleneck during FL. Moreover, due to the heterogeneity of the hardware, large models may be hard to train on small devices. Due to that, there are several recent approaches that aim to minimize communication via methods of quantization, which also aim to allow for smaller models via methods of compression and model quantization.

In this paper, the authors suggest a combination of two methods to reduce communication and allow for large model training by 1) using a lossy compressed model when that is communicated from the cloud to the edge devices, and 2) subsampling the gradients, a form of dropout, at the edge device side that allows for an overall smaller model update. The novelty of either of those techniques is quite limited as individually they have been suggested before, but the combination of both of them is interesting.

The paper is overall well written, however there are two aspects that make the contribution lacking in novelty. First of all, the presented methods are a combination of existing techniques, that although interesting to combine together, are neither theoretically analyzed nor extensively tested. The model/update quantization technique has been used in the past extensively [eg 1-3]. Then, the “federated dropout” can be seen as a “coordinate descent” type of a technique, i.e., randomly zeroing out gradient elements per iteration.

Since this is a more experimental paper, the setup tested is quite limited in its comparisons. For example, one would expect to see extensive comparisons with methods for quantizing gradients, eg QSGD, or Terngrad, and combinations of that with DeepCompression. Although the authors do make an effort to experiment with a different set of hyperparameters (dropout probability, quantization levels, etc), a comparison with state of the art methods is lacking.

Overall, although the combination of the presented ideas has some merit, the lack of extensive experiments that would compare it with the state of the art is not convincing, and the overall effectiveness of this method is unclear at this point.

[1] https://arxiv.org/pdf/1510.00149.pdf
[2] https://arxiv.org/pdf/1803.03383.pdf
[4] https://arxiv.org/pdf/1610.05492.pdf

---

> ### Author Response · Authors · 2018-11-19
> **We thank the reviewer for their thorough review. However, we think the review does not fully recognize the challenges of FL (Part 2)**
>
> The second point we want to address is our lack of comparisons against previous existing work:
>
> 1) Comparison with QSGD or Terngrad: We did not compare with these for two reasons.
> a) These methods were proposed for compression of gradient updates. In particular, the Terngrad paper argues for using the empirical distributions of the coefficients of such gradients. Even though those arguments would not directly apply to our setting, we could probably still use it for the Client-to-Server compression. However, we do not see a good reason why the proposal would be useful for compressing the state of the model being trained (i.e. Server-to-Client), which is the central concern of our paper.
> b) We performed a series of preliminary experiments where we compressed a variety of random vectors using QSGD and other techniques. The results of these small experiments suggested that in the tradeoff between accuracy and representation size, (I) uniform quantization was dominated by QSGD, and (II) QSGD was in turn dominated by the combination of Kashin’s representation and uniform quantization.
>
> We are happy to improve our Related Work section but, unfortunately, the rebuttal period will not be enough to fully recreate experiments using QSGD and Terngrad. What we could do in the time given is add the results of the simple experiments we mention above. We thus ask the reviewer, in light of our previous reasoning and the findings of our preliminary results, whether they consider the full comparison necessary, or whether adding the simpler evaluation would be sufficient.
>
> 2) Comparison with HALP: As far as we can see, the ideas introduced in that paper are largely compatible with our proposed methods (particularly regarding how we compute gradients locally) but would not replace them. We were previously unaware of this paper though, and we will add an appropriate reference to it.
>
> 3) Comparison with https://arxiv.org/abs/1610.05492: We clearly call out that we build on that work, and extend in two significant aspects. First, we introduce the use of Kashin’s representation (novel in ML in general) to further improve efficiency of uniform quantization. Second, we show how we can use the techniques in reducing Server-to-Client communication as well.

---

> ### Author Response · Authors · 2018-11-19
> **We thank the reviewer for their thorough review. However, we think the review does not fully recognize the challenges of FL (Part 1)**
>
> We thank the reviewer for their thorough review and for highlighting that the paper is well written. However, we think the review does not fully recognize the challenges of FL, and consequently misunderstands the nature (and therefore novelty) of our techniques. Please see a detailed explanation below.
>
> The first point we want to address is the (reported lack of) novelty of the following two contributions:
> 1) The lossy compression of the model sent from server to clients (the review points to other related works).
> 2) Federated Dropout, which the review mentions can be seen as a “‘coordinate descent’ type of a technique”.
>
> Let us address the two in turn:
> 1) We are not aware of previous work (and please correct us if we have missed something) that compresses the *state of a model* being trained when such compression has to be done repeatedly throughout the iterative training procedure and in a data-independent fashion. Techniques such as DeepCompression modify the whole training procedure, are data dependent, and produce one final compact model (i.e. compression is performed once). As such, not only do they become infeasible in the setting of FL (no data is available on the server), but they are not directly comparable with our method. Note that we do call this out in the last paragraph of Section 2 in the original submission, and highlight it could be *compatible* with the overall objective of FL. A proper exploration of such an idea, however, would likely deserve a complete paper.
> Furthermore, the idea of using Kashin’s representation can be of independent interest. We are not aware of any example of this idea being practically used in Machine Learning and, in the Appendix, we show its relationship to some recent theoretical results.
>
> 2) Claiming that Federated Dropout can be seen as coordinate descent, or that it can be reduced to subsampling gradients, is incorrect. In each client, we are not computing partial derivatives of the global model, but the full gradients of a smaller, and different, model. Furthermore, several SGD steps are taken for each local model. The facts that (a) by design of the procedure, we can then map these updates to the larger global model, and that (b) performing training this way leads to savings both in communication and local computation, are our key insights. We are not aware of this conceptual idea being addressed in previous literature. Finally, we do (optionally) use subsampling to further compress the final learned updates (together with basis transform and quantization), but this is complementary to (and not equivalent to) Federated Dropout.
>
> In summary, we believe that not only is the combination of our techniques interesting (as the reviewer points out), but that each individual technique does indeed bring novel ideas that address challenges where there is no state of the art at all.

---

### Author Response · Authors · 2018-11-19
**We thank all reviewers for their suggestions. We think a common misunderstanding among the reviews is that they don’t fully recognize some aspects of Federated Learning.**

We thank all reviewers for their suggestions and helping us see how the paper can be improved.

We think the common misunderstanding among the reviews is that they don’t fully recognize some aspects and challenges of Federated Learning (FL). We provide individual responses of why some of the reviewers’ suggestions are infeasible in FL, and explain other concerns.

In addition, we have discovered a minor flaw in how we explained Federated Dropout in the context of convolutional layers (unnoticed by the reviewers).  Additional change: We have made an improvement with respect to Federated Dropout applied to convolutional layers. Previously, we used it similarly as in the standard dropout, which did not let us realize space savings. In the updated version, we drop whole filters, which leads to both computational and communication savings. We repeated the experiments, and the conclusions still hold.

We thank all the reviewers for their comments highlighting the paper is overall well written!

---

### Meta-Review · Area_Chair1 · 2018-12-16
**A well-written paper addressing an important problem, but somewhat limited novelty and empirical evaluation**

**Confidence:** 4
**Recommendation:** Reject

**Metareview:**

This paper focuses on  communication efficient Federated Learning (FL) and proposes an approach for  training  large models on heterogeneous edge devices.   The paper is well-written and the approach is promising, but all reviewers pointed out that both novelty of the approach and empirical evaluation, including comparison with state-of-art, are somewhat limited. We hope that suggestions provided by the reviewers will be helpful for extending and improving this work.